# The α-Ketoglutarate Dehydrogenase Complex as a Hub of Plasticity in Neurodegeneration and Regeneration

**DOI:** 10.3390/ijms232012403

**Published:** 2022-10-17

**Authors:** Grace E. Hansen, Gary E. Gibson

**Affiliations:** 1Department of Biology, University of Massachusetts, Lowell, MA 01852, USA; 2Weill Cornell Medicine, Brain and Mind Research Institute, Burke Neurological Institute, White Plains, NY 10605, USA

**Keywords:** α-ketoglutarate dehydrogenase complex, metabolic plasticity, mitochondria, Alzheimer’s disease, tricarboxylic acid cycle, cell signaling, oxidative stress, transcription, α-ketoglutarate

## Abstract

Abnormal glucose metabolism is central to neurodegeneration, and considerable evidence suggests that abnormalities in key enzymes of the tricarboxylic acid (TCA) cycle underlie the metabolic deficits. Significant recent advances in the role of metabolism in cancer provide new insight that facilitates our understanding of the role of metabolism in neurodegeneration. Research indicates that the rate-limiting step of the TCA cycle, the α-ketoglutarate dehydrogenase complex (KGDHC) and its substrate alpha ketoglutarate (KG), serve as a signaling hub that regulates multiple cellular processes: (1) is the rate-limiting step of the TCA cycle, (2) is sensitive to reactive oxygen species (ROS) and produces ROS, (3) determines whether KG is used for energy or synthesis of compounds to support growth, (4) regulates the cellular responses to hypoxia, (5) controls the post-translational modification of hundreds of cell proteins in the mitochondria, cytosol, and nucleus through succinylation, (6) controls critical aspects of transcription, (7) modulates protein signaling within cells, and (8) modulates cellular calcium. The primary focus of this review is to understand how reductions in KGDHC are translated to pathologically important changes that underlie both neurodegeneration and cancer. An understanding of each role is necessary to develop new therapeutic strategies to treat neurodegenerative disease.

## 1. Introduction—An Overview of the Multiple Roles of KGDHC and the TCA Cycle

Exciting new research demonstrates how metabolism and mitochondria are central to processes causing neurodegeneration. Stimulating new research on metabolism in cancer suggests multiple new ways to approach the dynamic role of metabolism in neurodegenerative processes. To reveal how transformative these findings are, we first present an overview of the role of the α-ketoglutarate dehydrogenase complex (KGDHC). KGDHC regulation of the tricarboxylic acid (TCA) cycle places it central to neurodegeneration and cancer [1,2,3] (Section 1). Decades of elegant research on the structure and function of this beautiful complex reveal the subunits of the complex, the precise amino acids in each subunit that perform each step of the catalytic reaction as well as demonstrate precisely where modifiers affect the enzyme complex [4] (Section 2). These fundamentals stimulated research to demonstrate the role of KGDHC activity in multiple neurodegenerative diseases, especially Alzheimer’s Disease, as well as in oncogenesis (Section 3). To understand the role in neurodegenerative disease one must understand the enrichment of KGDHC activity in different cell types within the brain (Section 4). Once these fundamentals of KGDHC in multiple cells were known, cell biologists realized that only thinking of KGDHC in the production of NADH and ATP was too narrow because KGDHC has multiple roles in cells. KGDHC controls the production of reactive oxygen species (ROS) as signaling molecules and can act as an antioxidant (Section 5). The activity of KGDHC determines whether the TCA cycle acts as an oxidative pathway to produce reducing equivalents or as a reductive pathway to produce molecules for biosynthesis (Section 6). The activity of KGDHC controls the levels of α-ketoglutarate (KG), which has implications for processes in the mitochondria, cytosol, and nucleus (Section 7). Additionally, KGDHC regulates the post-translational modification of hundreds of proteins through succinylation (Section 8). KGDHC also controls transcription through succinylation and by regulating demethylation (Section 9). KGDHC controls cellular protein signaling (Section 10), and calcium in multiple cellular compartments (Section 11). An understanding of the multifaceted actions of KGDHC is necessary to develop new therapeutic approaches for both neurodegeneration and cancer.

The comparison of the rate of metabolism through the TCA cycle (i.e., flux) and KGDHC activity suggests that the KGDHC is rate controlling for the whole TCA cycle [1,2,3] (Figure 1). Glucose-derived pyruvate is decarboxylated by the pyruvate dehydrogenase complex (PDHC) to acetyl CoA, which combines with oxaloacetate to form citrate and to initiate the tricarboxylic acid (TCA) cycle [1,2,3]. The TCA cycle reduces NAD^+^ to NADH, FAD^+^ to FADH_2_, and produces GTP. NADH and FADH_2_ pass their electrons to the protein complexes of the electron transport chain to generate ATP. However, the NADH produced from KGDHC can be directed to the enzyme nicotinamide nucleotide transhydrogenase (NNT) to transfer hydride ions to NADH forming NADPH [5]. This is significant because NADPH is critical for maintaining the redox state of the cell by reducing glutathione [6]. In addition, the NADPH is critical for the reductive carboxylation of KGDHC, which is important for metabolic plasticity and oncogenesis (Section 6). Overexpression of NNT is sufficient to stimulate glutamine oxidation and reductive carboxylation, whereas it inhibits glucose catabolism in the TCA cycle [7].

## 2. Structure and Function of KGDHC

KGDHC catalyzes the fourth step of the TCA cycle and converts α-ketoglutarate (KG) to succinyl CoA using coenzyme A. The reaction releases CO_2_ and reduces NAD^+^ to NADH (Figure 2).

The enzyme complex consists of multiple subunits of three enzymes: α-ketoglutarate dehydrogenase (E1k), dihydrolipoamide succinyltransferase (E2k), and dihydrolipoamide dehydrogenase (E3) (Figure 2, Appendix A). In some tissues, oxoglutarate dehydrogenase L (E1k-L) substitutes for E1k [8]. The three subunits are arranged in an octahedral shape where multiple E2k proteins form the inner core and multiple E1k and E3 subunits are non-covalently bonded to the outside of the core [9,10,11,12,13,14] (Figure 2). Important molecular details of each protein and their regulators are presented in Appendix A. Interactions between the respective E1k and E2k components are typically strong, whereas E3 binds relatively weakly to the E2 subunits [15,16,17]. In E-Coli, the subunit composition of the native KGDHC is about 12 E2k chains, 24 E2k chains, and 12 E3 chains (Appendix A) [9]. Different chain ratios (E1k:E2k:E3) have been reported in the literature for mammalian KGDHC including 1:1:1.5 [18], 1:1.2:1.4 [19], and 1:1:0.5 [20,21]. The variations of the composition have been reviewed [10]. Compared with the heart complex, KGDHC isolated from the brain has an excess of the first component over the second and third. Data obtained using the three independent approaches suggest differences in the supramolecular organization of KGDHC from the heart and brain [19]. The significance of these differences in protein composition remains to be discovered.

The structure and function of the three proteins of KGDHC have been reviewed in detail [22] (Appendix A). While the subunit proteins are mostly considered for their role in the complex, they seem to have functions independent of complex activity for the control of oxidants in cells [23] and the regulation of post-translational modifications [24]. Each subunit of KGDHC has unique cofactors, activators, and inhibitors [25] (Appendix A).

The first subunit is the E1k protein, α-ketoglutarate dehydrogenase (KGDH). E1k catalyzes two reactions:$$\left(ThDP-E1\right)+\left(COOH-RC=O\right)\to \left(CO_{2}\right)+\left(OH-RC-ThDP-E1\right)$$
(OH−RC−ThDP−E1)+ (lip(S−S)−E2)→(ThDP−E1)+ (O=RC−S−lip(SH)−E2)

E1k utilizes thiamine diphosphate (TPP) as a cofactor. KG requires thiamine pyrophosphate (TPP) before binding to the E1 subunit. Once TPP and KG have formed a decarboxylated-generating hydroxyethyl compound bound to E1k, TPP transfers an acetyl group to lipoic acid (LA) [26]. The TPP is tightly but not covalently bound to E1k. E1k catalyzes the oxidative decarboxylation of α-ketoglutarate, and the subsequent binding of the resulting succinic acid fragment to a sulfur residue of lipoic acid on the second enzyme E2k, with concomitant regeneration of the TDP on E1k. This reaction is physiologically irreversible, due to the diffusion away of the CO2 product. E1k is rate limiting for the complex and is affected by the majority of the regulators of the overall process, including allosteric responses to second messengers and metabolic indicators, such as Ca^2+^, ATP/ADP, SH/-S-S (thiol/disulfide), NADH/NAD+, and acyl-CoA/CoA [27]. The kinetic properties of E1k-like proteins have the lowest catalytic activity (‘turnover number’) among the components and as such limits the rate of the overall process. Since E1k is rate limiting, many of the regulators of the overall reaction act on E1k [28]. Further E1k expression is often increased in response to the KGDHC impairments [27].

A relatively recently discovered brain-specific KGDHC subunit, α-Ketoglutarate dehydrogenase-like (KGDHL) may be particularly critical in neurodegeneration. Its properties have been reviewed in detail (Appendix A) [8]. Sequence analysis indicates that KGDHL represents a previously unknown isoform of KGDH [8]. Conservation of all the essential residues confirmed their catalytic competence. The signatures of the Ca^2+^-binding sites were found in the Ca^2+^-activated KGDH and KGDHL [8]. KGDHL occurs in the brain, but not the heart [19].
$$\left(O=RC-S-lip\left(SH\right)-E2\right)+\;\left(HS-CoA\right)↔\left(O=RC-S-CoA\right)+\left(lip{\left(SH\right)}_{2}-E2\right)$$
$$\left(lip{\left(SH\right)}_{2}-E2\right)+\left(E3\left(S-S\right)-FAD\right)↔\left(lip\left(S-S\right)-E2\right)+\left(E3{\left(SH\right)}_{2}-FAD\right)$$

The core enzyme, E2k, catalyzes the transfer of the four carbon fragments from the TDP on E1k to coenzyme A, producing succinyl-CoA. It also transfers reducing equivalents to the flavoprotein moiety attached to the E3 protein. E2k contains one or more dihydrolipoic acid residues covalently bound in amide linkages to the amino groups of one or more lysines. In E2k, the thiol group of Coenzyme A takes the place of the thiol group on E2k through a transesterification reaction. The cofactors for E2k are lipoic acid and coenzyme A [29]. This reaction produces succinyl-CoA and dihydrolipoic acid (DHLA), the reduced form of LA [26]. The E2k subunit is inhibited by succinyl CoA and ATP.
$$\left(E3{\left(SH\right)}_{2}-FAD\right)+\left(NAD^{+}\right)↔\left(E3\left(S-S\right)-FAD\right)+NADH+H\;$$

E3 is responsible for oxidizing dihydrolipoic acid to lipoic acid for use in the reaction again. The cofactors for E3 are FADH_2_ and NAD^+^. E3 also reduces NAD^+^ to NADH and produces reactive oxygen species (ROS). The E3 subunit is inhibited by ATP and NADH [29]. E3 is shared by PDHC, branched-chain dehydrogenase complex, and glycine cleavage system [29].

## 3. Neurodegenerative Diseases with Disrupted KGDHC

Alterations of KGDHC activity have been implicated in multiple neurodegenerative diseases (Table 1). In infantile lactic acidosis, a mutation in the DLD or E3 subunit of KGDHC is associated with intense psychomotor retardation [30,31]. Disruption of E2k function has been linked to childhood psychomotor retardation [32,33]. Mutations in the DLD G229C gene of the E3 subunit occur in some intermittent neuropsychiatric diseases with ataxia and in children with attention deficit disorders as well as a lack of coordination [34]. Friedreich’s Ataxia, which causes damage to the long tracts of the spinal cord, is caused by a GAA repeat in the FRDA gene, which leads to a deficiency of the E3 subunit of KGDHC [35,36,37,38]. Further, reductions in KGDHC activity are implicated in Parkinson’s disease and may occur either indirectly or as the result of a change in the DLST gene. Parkinson’s disease includes a loss in motor function as well as a decline in cognitive function [39,40,41]. KGDHC activity is also diminished in Huntington’s Disease, which includes both motor and cognitive changes [42]. The decline in KGDHC activity in Alzheimer’s disease correlates to loss of cognitive function [36,41,42,43,44,45]. Reductions of KGDHC in the superior frontal cortex and cerebellum occur in progressive supranuclear palsy, which leads to postural instability and some cognitive impairment [46,47]. In addition to these neurodegenerative diseases, traumatic injuries to the spinal cord reduce KGDHC in the spinal cord by 90% and in the cerebellum by 30% [48]. The cortex also shows a reduction of KGDHC in animal models of head trauma [45]. KGDHC is also diminished in brains of patients with Wernicke-Korsakoff syndrome [45]. Thus, brain KGDHC activity is sensitive to many diseases and injuries. Therefore, understanding the wide variety of roles KGDHC plays in cells will have widespread implications for multiple diseases.

The role of KGDHC in disease has been widely studied in AD. This research was stimulated by the extensively replicated finding that glucose metabolism declines in AD in parallel with cognition [44] and that thiamine and thiamine-dependent enzymes are known to be linked to memory [45]. Thiamine deficiency leads to severe memory deficits like those in AD [50]. The activity of the major thiamine-linked enzymes (pyruvate dehydrogenase complex, transketolase, and KGDHC decline in AD [38,51,52,53,54,55,56]. The decline of KGDHC in AD parallels the reduction in cognitive function [51,54]. Some of the deficit is related to a lack of TPP since adding TPP to the assay mix enhances activity more in samples from AD than controls [56]. The reduction in thiamine diphosphate correlates strongly with brain glucose hypometabolism in Alzheimer’s disease, whereas amyloid deposition does not [57]. Whether the protein levels of each subunit change with AD varies between investigators and brain region [58]. In brains from patients bearing AD causing APP670/670 mutations, the levels of E1k and E2k but not E3 decline [53]. KGDHC activities are more markedly reduced than protein levels suggesting that decreased enzyme activity cannot be primarily explained by loss of KGDHC protein levels [38]. Thus, while the reduction in activity is well documented, neither the cause, nor the consequences, are established. Both oxidant modifications or glutathionylation have been suggested as causes as addressed in previous reviews [59]. The remainder of this review addresses how the reduction in KGDHC activity may be key to AD and the potential impact of targeting KGDHC for therapeutic treatments.

The use of transgenic mice in which KGDHC is genetically reduced is a powerful approach to studying the implications of KGDHC in disease. Genetic manipulations to reduce the activity of KGDHC subunits lead to specific AD-related changes (Table 2). For example, mice deficient in DLD have increased vulnerability to multiple metabolic toxins and diminished neurogenesis [60]. DLST deficiency accelerates amyloid pathology and memory deficiency in a transgenic mouse model of amyloid deposition [61]. E2k-controlled succinylation directly links KGDHC to plaque and tangle formation [24] (see Section 8). KGDHL declines in mouse models of AD. Genetically reducing KGDHL increases plaques and overexpression of KGDHL diminishes plaques [62]. Thus, multiple genetic animal models support the key role of KGDHC in AD.

Thiamine deficiency models the mild impairment of metabolism observed in AD. Thiamine deficiency diminishes the activity of KGDHC as well the thiamine dependent enzyme transketolase and causes selective neuron loss. Thiamine deficiency diminishes activity of KGDHC by 36% without altering PDHC activity. Treatment with thiamine for seven days reversed the neurological symptoms of AD and restored glucose oxidation and KGDHC activity [76]. In animal models, thiamine deficiency reduces KGDHC activity and causes selective neurodegeneration, AD-like changes in plaques, tangles, use of glucose and inflammation as well as neuron loss [69]. While thiamine deficiency causes AD like changes in animal models, pharmacological levels of thiamine diminish plaques [81] and tangles [82] in mouse models of AD pathology. Thus, in animal models, increasing thiamine levels increases KGDHC activity and reduces neurological symptoms characteristic of AD.

Patients with Alzheimer’s disease have a low expression of KGDHL, and research in animal models demonstrates the importance of these changes. Two-dimensional gel electrophoresis and mass spectrometry analysis indicate that KGDHL is reduced in 3 × Tg-AD [83]. Subsequently, the role of KGDHL was shown in several AD-associated phenotypes in mice via the AAV-PHP.eB-mediated upregulation of KGDHL. Increasing KGDHL improves the memory ability of triple-transgenic AD (3 × Tg-AD) mice and diminishes AD-like pathology. Overexpression of KGDHL activates the Wnt/b pathway by increasing the expression of Wnt7B and helps to maintain mitochondrial homeostasis [62].

## 4. KGDHC in Brain Regions and Brain Cell Types

The heterogeneity of KGDHC in various brain regions complicates the ability to make direct connections to the pathophysiology of AD and many other diseases. The activity in various brain regions varies by more than two-fold from 19.2 mU/mg protein in the cortex to 7.6 mU/mg protein in olfactory bulbs [84]. Within a brain region, the activity may vary between layers such as in the cortex. In the cortex, high immunoreactivity occurs mostly in layers III, V, and VI. The hippocampal pyramidal layer in CA1 and CA2 exhibits more intense staining than in CA3. In the mammillary body, intensely labeled cells occur in the supra-mammillary and lateral nuclei, while moderately stained cells predominate in the medial nucleus. The basal forebrain, basal ganglia, reticular and midline thalamic nuclei, red nucleus, pons, cranial nerve nuclei, inferior and superior colliculi, and cerebellar nuclei also contain highly immunoreactive neurons. In many brain regions, the distribution of KGDHC overlaps with that of PDHC and co-localizes to a limited extent with choline acetyltransferase [85]. Thiamine deficiency dramatically reduces KGDHC activity in both anatomically damaged (thalamus and inferior colliculus) and spared (cerebral cortex) regions. Immunocytochemistry reveals no apparent correlation in regional KGDHC immunoreactivity or its response to TD with affected regions in TD [86]. Although KGDHC activity declined during thiamine deficiency, the protein levels of the three subunits did not decline [86]. Comparison of activity in homogenates, also reveals no differences between mammillary bodies, inferior colliculi, or cochlear nuclei [87].

Regional heterogeneity of KGDHC is also observed in the human brain. KGDHC immunohistochemistry of autopsied temporal neocortex of non-AD cases reveals a distinctive laminar pattern of intensely immunoreactive cells in layers III and V. These KGDHC-immunoreactive cells show the characteristic somato-dendritic morphology of pyramidal neurons at higher magnifications. Much less marked immunoreactivity is seen in other cortical neurons. The KGDHC-positive neurons decline dramatically in the brains of AD patients [88,89].

In addition to differences in the regional heterogeneity of KGDHC, differences in cell types must be considered. Surprisingly, convincing immunoreactivity of E2k is not found in the glia in the human brain [88,89]. The replicated finding of low or missing E2k in human astrocytes occurs despite high KGDHC activities in rodent astrocytes. Indeed, astrocytes isolated from rodents have three times the KGDHC activity as neurons [90]. KGDHC is readily apparent in cultured astrocytes [91]. Both KGDHL and KGDH show a similar labeling pattern [89]. Immunochemical studies of the human cortex suggest KGDH, KGDHL, and DLST are mostly expressed in neurons (medium or high staining), and they exhibit a low expression in the glia. DLD staining is medium for both neurons and glia [89]. Mapping studies only measure protein levels, which do not necessarily reflect protein activity. Thus, major cell types were isolated and the KGDHC activities of each cell type were measured. The activity of KGDHC is uniquely distributed among different cell types (Figure 3). The highest activity is in the endothelial cells, and the lowest is In primary microglia and neurons [90]. Mitochondrial heterogeneity in various cell types in other organs has been extensively reviewed [92]. While the relative activities as compared to protein levels are very important in understanding the role of KGDHC in the pathophysiology of a disease, the studies described in Figure 3 have important limitations: 1. They have not been replicated. 2. Cells in culture are very different than in tissue. 3. The figure includes cells that have been transformed which alters metabolism.

A deficiency of thiamine, the precursor of the co-factor thiamine pyrophosphate, produces selective neuron death but activates other cell types (Figure 3). The unique distribution of KGDHC activity does not account for the selective response to thiamine deficiency. TD slightly inhibits general cellular dehydrogenases in all cell types, whereas it significantly reduces the activity of KGDHC exclusively in primary neurons. Among the cell types tested, only in neurons did thiamine deficiency induce apoptosis and cause the accumulation of 4, hydroxy-2-nonenal, a lipid peroxidation product [90]. Thus, in vivo and in vitro, thiamine deficiency kills neurons and activates other cell types [90].

## 5. KGDHC and Cell Signaling by Reactive Oxidative Species (ROS)

KGDHC is central to cell signaling regulated by reactive oxygen species (ROS). ROS regulates hundreds of cell processes. Uniquely, KGDHC is both regulated by ROS and produces ROS (Figure 4). The data strongly indicates that the E3 subunit of KGDHC is a primary site of

ROS production in normally functioning mitochondria [93,94]. When NADH levels are high, the E3 subunit of KGDHC is a major producer of ROS, specifically superoxide and hydrogen peroxide [95]. H_2_O_2_ production by brain mitochondria is significantly diminished in mitochondria isolated from heterozygous knock-out mice deficient in E3. At optimal conditions for each system, superoxide/H_2_O_2_ was produced by the KGDHC complex at about twice the rate from the PDHC complex, four times the rate from the branched chain ketoacid dehydrogenase complex, and eight times the rate of complex I [96]. Depending on the substrates present, KGDHC activity is decreased by oxidative stress [59] and chronic hypoxia [97]. The pattern of oxidant modification of KGDHC subunits and activity is similar to that seen in AD [59]. KGDHC is inactivated by a variety of oxidants including peroxynitrite, nitric oxide, hydroxylnonenal, H_2_O_2_, chloroamine, superoxide donor xanthine, hypochlorous acid, mono-N-chloramine, and sodium hypochlorite [59]. The inactivation can occur by multiple mechanisms. For example, peroxynitrate inhibits KGDHC by nitration of tyrosine residues. H_2_O_2_, tert-butylhydroperoxide and hydroxynonenal decrease KGDHC activity by oxidizing the lipoic acid on the E2 subunit [59,98]. In intact mitochondria, KGDHC inactivation by ROS can only take place in the presence of substrate [98]. Treatment of rat heart mitochondria with the oxidant hydroxynonel (HNE) selectively inhibits KGDHC and PDHC, while other NADH-linked dehydrogenases and electron chain complexes are not affected. The inactivation by HNE is greatly enhanced in the presence of substrates that reduce the sulfur atoms of lipoic acid covalently bound to the E2 subunits of KGDGC and PDHC. In addition, loss of enzyme activity induced by HNE correlated closely with a decrease in the availability of lipoic acid sulfhydryl groups [99]. With H_2_O_2_ treatment, the E2 subunit of KGDHC is reversibly glutathionylated. The lipoic acid, a required cofactor covalently attached to the E2 subunit is the site of glutathionylation. The relative level of glutathionylated lipoic acid closely paralleled the degree of enzyme inhibition and reactivation. Glutathionylation of KGDHC protects lipoic acid from modification by the electrophilic lipid peroxidation product 4-hydroxy-2-nonenal. Glutathionylation of KGDHC can therefore be viewed as an antioxidant response protecting the enzyme from oxidative damage from ROS (Figure 4) [99,100,101]. Simply removing ROS from an isolated enzyme does not reactivate the enzyme, indicating that there is some irreversible damage [59,98]. Glutathione reversibly inhibits KGDHC [23,101], and this protects KGDHC from oxidants. When ROS is no longer present, glutaredoxin can remove glutathione and reactivate the enzyme, protecting KGDHC from damage. This unique process protects KGDHC.

KGDHC can also act as an antioxidant as shown by the observation that reductions of either E2k or E3 increase oxidative stress [60]. One plausible mechanism is the link of KGDHC to nicotinamide nucleotide transhydrogenase (NNT) (Figure 1). NNT converts mitochondrial NADH to NADPH. NADPH is key to controlling the redox state of the cytosol and the nucleus. NADPH also reduces glutathione whose regulation of the cellular redox state is pervasive. NADH from KGDHC selectively shuttles to NNT for NADPH formation rather than to complex I of the respiratory chain for ATP production, which implicates this path in KGDHC’s role as an antioxidant in cells [5].

## 6. KGDHC and Metabolic Plasticity

KGDHC’s position at the intersection of multiple metabolic pathways makes it central to metabolic plasticity (Figure 5). KGDHC controls metabolic signaling in the mitochondria and between the mitochondria and cytosol and nucleus. This metabolic plasticity is utilized by cancer cells to maintain high levels of growth and proliferation by reprogramming mitochondrial metabolism. This high level of proliferation requires citrate, but in the normal oxidative pathway of the TCA cycle, citrate is broken down. Cancer cells can change this cycle by inhibiting KGDHC which produces a large NADH/NAD+ ratio and leads to a higher KG to succinate ratio in the mitochondria [14]. The excess KG that can no longer be broken down by KGDHC stimulates the reverse reaction or the reductive pathway of the TCA cycle. KG is converted to isocitrate and then citrate to promote fatty acid synthesis (Figure 5) to support tumor growth [15].

Metabolomics flux analysis reveals enhanced reductive carboxylation upon genetic deletion of the E2 subunit of KGDHC (DLST) mimicking pharmacological inhibition of the complex [102]. The reduced KGDHC activity in AD may have a similar consequence. Lipid droplets are universal in Alzheimer’s disease and occur early in the disease process. Lipid droplets are evolutionarily conserved organelles that dynamically stockpile fatty acids [103]. The data are consistent with the suggestion that the AD-related decline in KGDHC promotes the reductive pathway, which results in enhanced citrate formation and fatty acid production.

Large amounts of KG are produced in the cell through glutaminolysis. Glutamine is a widely available amino acid that is an important source of nitrogen and carbon for rapidly growing cells [104]. In cancer cells, glutaminolysis occurs at extremely high levels producing a large amount of KG [105]. If KGDHC is inhibited, accumulated KG will be directed through the reverse steps of the TCA cycle, which leads to more citrate for lipid synthesis in cancer cells. Hypoxia is an important component of cancer and neurodegeneration. Under hypoxic conditions, KG production is increased which leads to more citrate production [106,107].

In cancer, KGDHL acts as a tumor suppressor. The downregulation of this enzyme contributes to the promotion of several different cancers. Cancer cells in the liver that have inhibited KGDHL complex have a high ratio of KG to citrate which causes the reductive pathway of the TCA and leads to lipid synthesis, providing fuel for growing cancer cells (see Section 7). When KGDHL is increased in cancer cells, there is also an increase in ROS production which leads to apoptosis of cancer cells. Additionally, the overexpression of KGDHL restricts cancer cell migration and invasion [108].

Relatively few papers have examined reductive carboxylation in the brain. It is likely important in development and neurodegeneration. Growing evidence for the significance of metabolism in regulating cell fate suggests understanding the metabolic programs may provide novel therapeutic approaches to address brain diseases. The proliferation and differentiation of neural stem cells (NSCs) have a crucial role to ensure neurogenesis and gliogenesis in the mammalian brain. Dramatic reductions in reductive carboxylation occur as neural stem cells are converted to astrocytes. NSCs consume glutamine from the medium; the highly active reductive carboxylation of KG indicates that this was converted to citrate and used for biosynthetic purposes. In astrocytes, pyruvate enters the TCA cycle mostly through pyruvate carboxylase (81%). This pathway supports glutamine and citrate secretion, recapitulating well-described metabolic features of these cells in vivo [109].

Furthermore, KG is both metabolically as well as structurally comparable to oncometabolites, metabolites that are present in high levels in cancer cells compared to normal cells. KG can regulate over 60 enzymes involved in a wide range of functions from fatty acid metabolism to epigenome editing. KG can act as a tumor suppressor, but it can also stimulate oncogenic functions. Understanding of the link between KG and cancer is rapidly expanding [108,110,111].

The wide scope of KG actions raises the question of whether KGDHC controls KG levels. It has directly been demonstrated that inhibiting KGDHC increases KG levels. Measurements of KG after inhibition of KGDHC demonstrate a 50–60 percent to four-fold increase in KG levels depending on concentration and timing of inhibition [112,113]. Similar increases in KG following inhibition of KGDHC occur in other cell models [114]. Cellular compartmentation suggests the increase may be much greater in some compartments. The consequences of KGHDC inhibition on flux were determined. Flux measures with [1-^13^C]-glucose and [U-^13^C] glutamate indicate that inhibition of KGDHC with specific KGDHC inhibitors in intact cerebellar granule neurons alters metabolism. The findings show that decreased KGDHC activity leads to increased KG levels and increased transamination of KG with valine, leucine, and GABA [115].

## 7. Interactions of KG and KGDHC Activities with HIF

KGDHC and KG may also regulate cell metabolism and function through an interaction with Hypoxia Inducible Factor 1 (HIF1), a protein complex that is responsible (Figure 6). The α-subunit contains an oxygen-dependent degradation (ODD) domain, which is hydroxylated by a KG-dependent proline-hydroxylase-2 (PHD-2), rendering the α-subunit vulnerable to proteasomal degradation under normoxic cellular conditions [116]. HIF is controlled by KG levels through PHDs. During normoxia, KG activates PHD, which blocks the dimerization of HIF, ensuring HIF is continuously degraded. Additionally, in normoxia, glutamine is converted to KG in the mitochondria and enters the TCA cycle for oxidation. During hypoxia, KG levels in the cytosol are limited, which leads to the inactivation of PHD and thus the stabilization of HIF. HIF activation promotes targeted ubiquitination by the three ubiquitin-protein ligase SIAH2 of the E1 subunit of KGDHC and subsequent proteolysis, inactivating KGDHC up to 60% [31,117]. This inactivation leads to the accumulation of KG in the mitochondria [112,113]. Since KGDHC is inhibited, KG accumulates and enters the reductive pathway of the TCA cycle to produce citrate to promote lipid synthesis. In addition, the KG produced from glutamine in the mitochondria is exported into the cytosol where it activates PHD to destabilize HIF (Figure 6).

Hypoxia inhibits KGDHC activity through SIAH2. Under hypoxia, there is no oxygen present to activate PHDs, which results in the stabilization of HIF. HIF activates the E3 ubiquitin-protein ligase SIAH2 which inhibits KGDHC. Since KGDHC is inhibited, KG cannot be oxidized into succinyl-CoA. Instead, KG will enter the reductive pathway of the TCA cycle, promoting lipid synthesis. 

Hypoxia Results in Accumulation of KG Which Reactivates PHD. With KGDHC inhibited, KG will begin to accumulate in the mitochondria. The KG that is produced through glutaminolysis leaves the mitochondrial and enters the cytosol. This KG activates PHD in the cytosol which then destabilizes HIF. So even under hypoxia, KGDHC regulation plays a role in response to hypoxia. 

Accumulation of KG prevents the activation of HIF1 by succinate, fumarate, and hypoxia. This reverses cell death and enhances glycolysis [118,119]. This demonstrates how the levels of KG can increase PHD activity as well as decrease the effects of hypoxia on HIF. Disruption in the HIF1 communication pathway underlies the antitumorigenic effects of KG [120]. KG antagonizes other oncometabolites giving it some tumor suppressor characteristics [120].

Thus, in cancer cells, a clear feedback loop exists between KGDHC and HIF. This circuit is necessary for the growth of cells in hypoxia and models of tumor formation but has not been explored in neurodegeneration. Not only does the inhibition of KGDHC control the activity of HIF through KG levels, but HIF controls the inhibition of KGDH through SIAH. Active HIF increases SIAH2 which ubiquitinates KGDHC and limits its activity up to 60% [117]. Inhibition of KGDHC produces an accumulation of KG which in turn will activate PHDs and destabilize HIF. This is a clear feedback loop in the cell that could be a very important therapeutic target.

## 8. KGDHC and Signaling by Post Translational Modification through Succinylation

Post-translational modifications of proteins by succinylation are regulated by KGDHC and provide an efficient and rapid biological regulatory mechanism that links metabolism to protein and cell functions. Traditionally, changes in KGDHC were thought to be transferred to cell function by changes in ATP/ADP, GTP, FADH_2_, and NADH. KGDHC is now known to cause changes in cell function by controlling the very precise post-translational modification succinylation. Succinylation of lysine groups induces more substantial changes to a protein’s chemical properties than does lysine acetylation or methylation, as it adds a bigger structural moiety and changes the charge from a positive to a negative charge [121]. The second protein of the KGDHC complex (E2k) is a succinyl transferase and the final product of the complex is succinyl-CoA. KGDHC controls succinylation throughout the cell. Convincing results in yeast reveal that induction of E1k increases cellular succinylation by about 1.7-fold and mitochondrial succinylation by 2.7- to 4.7-fold. Loss of E1k causes a four-fold reduction in cellular succinylation and a six-fold reduction in mitochondrial succinylation [122].

Multiple experiments in cells support the hypothesis that succinylation links altered metabolism to multiple cellular functions. Changes in succinylation of mitochondrial proteins following variations in metabolism were compared against the mitochondrial redox state as estimated by the mitochondrial NAD^+^/NADH ratio. Multiple metabolic perturbations including reduced glycolysis, glutathione depletion, depressed TCA cycle activity, impairment of electron transport, ATP synthase, and uncouplers of oxidative phosphorylation dramatically decreased succinylation. In contrast, reducing the oxygen from 20% to 2.4% increased succinylation. The results demonstrate that succinylation varies with metabolic states, is not correlated to the mitochondrial NAD^+^/NADH ratio, and appears to coordinate the response to metabolic challenges [123].

KGDHC also controls succinylation in neuronal cell models [59,80] (Figure 7). Reduction of KGDHC activity with a very specific inhibitor diminishes succinylation of both cytosolic and mitochondrial proteins in cultured mouse neurons and in a neuronal cell line. The inhibitor caused a concentration-dependent inhibition of in situ KGDHC activity by 60% in N2a cells and 35% in primary cultured cortical neurons [65]. A structural analog that did not inhibit KGDHC did not alter succinylation. The reduction of succinylation in the mitochondria is nearly complete. Purified KGDHC can succinylate multiple proteins including other enzymes of the TCA cycle leading to modification of their activity. The much greater effectiveness of succinylation by KGDHC than succinyl-CoA suggests that the catalysis by the E2k succinyltransferase is important. Inhibition of KGDHC also modifies acetylation by modifying the PDHC. Thus, succinylation appears to be a major signaling system that is regulated by KGDHC [59,80].

The human brain contains hundreds of succinylated proteins [24]. A succinylome of twenty human brains revealed 1908 succinylated peptides from 314 unique proteins [24]. While succinylation in the mitochondria is most prominent (229 proteins), succinylation is also common in the cytosol (95 proteins), the nucleus (73 proteins), and the plasma membrane (59 proteins), with fewer in multiple other compartments. The number of succinylated sites per protein varies from one to greater than ten [24]. While the functional significance of most of these modifications remains largely unknown, each enzyme that has been studied shows that succinylation is physiologically and pathologically important [80].

Reductions in KGDHC activity have been reported in numerous diseases (see Table 1). Since succinylation plays a role in regulating many metabolic pathways, the decline in KGDHC activity may alter cell function through succinylation (Figure 8). This interaction has only been studied in AD. While succinylation of mitochondrial proteins decreases in AD, the succinylation of cytosolic proteins increases. The data suggest that the increased succinylation of cytosolic proteins is due to the translocation of KGDHC from the mitochondria to the cytosol [24]. Similarly, following intracerebral hemorrhaging mitochondrial protein succinylation decreases while cytosolic protein succinylation increases [124]. The studies suggest that KGDHC leaves the nucleus and migrates to the cytosol to succinylate cytosolic proteins [24].

The changes in succinylation with AD are very specific and are closely linked to the development of the pathology [24]. A comparison of ten AD and ten control brains revealed a significant decline in the succinylation in 19 mitochondrial proteins while the succinylation of 12 cytosolic proteins increased [24]. In AD, tau succinylation increased at the “PHF 6 sequence”. Experiments with isolated proteins showed that succinylation of this sequence blocks the binding of tau to microtubules and promotes tau aggregation. In AD, succinylation of APP increased at the lysine in α secretase. Experiments with isolated proteins show succinylation blocks α secretase and promote beta-amyloid production and amyloid aggregation. Thus, succinylation directly links metabolic changes to the major attributes of AD: amyloid, tau, and neurodegeneration [24].

Our understanding of the interactions of various post-translational modifications is in its infancy. Since both acetylation and succinylation modify lysine groups, many proteins can be modified at the same sites [122]. The interactions of phosphorylation, succinylation, and acetylation have also been studied at the protein levels. The three post-translational modifications have similar effects on tau binding to the microtubule. Thus, these post-translational modifications may influence both the physiological and pathological interactions of tau and thus represent targets for therapeutic intervention [125]. Both modifications link metabolic changes to changes in function [126]. The interactions are further complicated by PDHC-mediated acetylation modifying KGDHC [127] and KGDHC succinylation modifying PDHC [80].

## 9. Nuclear KGDHC Alters Transcription (Epigenetic Effect)

The signaling by KGDHC and KG through succinylation extends to the nucleus (Figure 9). The succinylomics analysis discussed above revealed multiple succinylated proteins in the nucleus [24]. Histone lysine succinylation (Ksucc) is common in the nucleus. Histone succinylation is regulated by KGDHC. In glioblastoma cells, a fraction of KGDHC is distributed in the nucleus in a manner that requires the presence of the nuclear localization signal in DLST. In the nucleus, KGDH interacts with the histone acetyltransferase 2A (KAT2A) at gene promoters and locally supplies succinyl-CoA, which is then bound by KAT2A with high affinity. Crystal structure analysis reveals the precise molecular sites that are important in the selective binding of KAT2A to succinyl-CoA. The high binding affinity of succinyl-CoA for KAT2A and the high local concentrations of succinyl-CoA generated by the KAT2A-associated KGDHC complex facilitate histone succinylation on K79 in the promoter regions of more than 7000 genes despite low concentration of succinyl-CoA in the nucleus. Inhibition of nuclear entrance of the KGDHC complex or expression of the KAT2A mutant with low binding affinity for succinyl-CoA reduces gene expression and inhibits tumor cell proliferation and glioma growth in mice [128]. These findings revealed that local generation of succinyl-CoA by the nuclear KGDHC coupled with the succinyltransferase activity of KAT2A is instrumental in histone succinylation and has a large impact on gene expression, particularly in the context of cancer, fueling tumor cell proliferation and tumor development [129]. Defects in the TCA cycle by the depletion of SDH increase succinyl-CoA, and subsequent histone hypersuccinylation correlates with active gene expression [130]. Chromatin succinylation correlates with active gene expression and is perturbed by defective TCA cycle metabolism [130]. Nuclear succinylation is also regulated by KGDHC. Knockdown of KGDHC in cultured myocytes reduces chromatin-bound KGDHC and succinylation at specific sites [131].

The TCA cycle enzymes in the mitochondria are also present in the nucleus [132] and are a part of an incomplete cycle that produces metabolic intermediates which control transcription by epigenetic modification. All the TCA mitochondrial enzymes except succinate dehydrogenase are found in the nucleus. In the nucleus, KGDHC is used in an incomplete cycle where the main objective is to produce intermediates for epigenetic regulation [132]. The molecular mechanism for regulation and how this localization is altered in AD is still unknown.

Both the enzyme KGDHC and the substrate KG are involved in epigenetics. Tumorigenesis can be altered by epigenetics at the histone level as well as the DNA level [133]. The varying cytosolic levels of KG are responsible for the methylation of DNA and histones which lead to changes in epigenetics. Histone demethylases use KG as a cofactor to facilitate the removal of methyl groups from histones. If KGDHC is inhibited, more KG accumulates in the cell, leading to higher levels of demethylases, and altering epigenetics [102,134,135]. Thus, sustaining a balanced ratio of KG to succinate is crucial to determining the destiny of an embryonic stem cell [136]. If the ratio is high, chromatins will become modified, and histones will become demethylated. Further, KG-dependent dioxygenase activity responds to the disruption of cellular oxygen levels and the changes in metabolism resulting in a closed chromatin structure as well as a decrease in gene expression [137,138,139].

One example of how KGDHC and KG regulation of gene expression may be important is in the regulation of autophagy and apoptosis by p53. In cells with low KG, activation of P53 induces ATG gene expression and apoptosis. Reduced autophagy-related gene expression contributes to P53-induced apoptosis. Addition of KG either by reducing KGDHC or adding cell permeable KG overcomes the P53 induction of apoptosis. Reduced levels of KG increase apoptosis and decrease autophagy while increased levels of KG promote apoptosis resistance and increase autophagy [140].

## 10. KGDHC and Cellular Protein Signaling

KGDHC has been implicated in protein signaling through its ability to induce the release of mitochondrial proteins into the cytosol (Figure 10). Reductions in KGDHC activity by 63–80 percent do not cause energy failure but do increase cytochrome c release [141,142]. In addition to cytochrome c, other mitochondrial proteins are released in response to impaired KGDHC activity. In metabolically compromised mitochondria, the DLST subunit of KGDHC leaves the mitochondria and enters the cytosol [24]. The data is consistent with this increased cytosolic KGDHC causing succinylation of unique sites on specific proteins including tau and APP in the cytosol [24]. Additional reports suggest the migration of mitochondrial proteins from metabolically compromised mitochondria to the cytosol. For example, mitochondrial aspartate aminotransferase leaves the mitochondria with anoxia [143,144]. The ability of these proteins to leave mitochondria likely depends on their mitochondrial localization [145].

The capability of mitochondria to control complex cellular protein signaling with the cytosol is well documented in regard to apoptosis (for review see [146]). Protein signaling modulates mitochondrial fusion and fission, which are necessary for mitochondrial health. Protein signaling is move sensitive to impaired KGDHC activity than are classical measures of energy metabolism such as ATP. Reduction of KGDHC activity with a very specific inhibitor of KGDHC, carboxyl ethyl ester of succinyl phosphate (CESP), alters protein signaling in the human neuroblastoma cell line SY5Y [147]. CESP reduces the activity of KGDHC by 23–53% over time without altering the mitochondrial membrane potential or ATP levels. Thus, this degree of inhibition of KGDHC does not cause a general failure of energy metabolism or cell death. Instead, inhibition of KGDHC causes translocation of Drp1 and LC3 from the cytosol into the mitochondria. In the mitochondria, Drp1 arranges into helical structures which induce mitochondrial fission where the mitochondria split into two smaller mitochondria [148]. The reduction of KGDHC activity increases cytochrome c release in parallel with the movement of Drp1 [141,142]. The movement of the cytosolic protein Bax to the mitochondria is also required for cytochrome c release. Bax binds to the mitochondria at the initial stages of the cell response. However, once the mitochondria divide, Bax has a difficult time binding to fragmented mitochondria which decreases cytochrome c release (Figure 9). Overall, analysis of LC3 and Drp1 by Western Blot as well as the change in mitochondrial shape demonstrate that a decreased KGDHC activity promotes mitophagy. The findings suggest that the regulation of mitochondrial dynamism by the intracellular KGDHC activity by recruitment of different players of autophagy/mitophagy, and altered protein signaling, may be critical in the neurodegenerative processes [73].

The consequences of KGDHC inhibition on protein signaling may be mediated by alterations in surface charge on the mitochondria. Inhibition of KGDHC alters the surface charge of the mitochondria which can alter protein signaling. Increased annexin V fluorescein following treatment of cells with CESP for one hour indicates a change in surface charge which corresponds to changes in mitochondrial signaling [73,149].

Studies of reduced KGDHC activity inducing mitophagy support the suggestion that diminished KGDHC activity can be protective in the short-term. Hence, increased fission and mitophagy induced by mild impairment of mitochondrial function is protective in the short-term, but ultimately makes the cells more sensitive to other conditions that impair metabolism. Other studies demonstrate that a short-term reduction in KGDHC activity causes a different response than long-term or chronic reductions in both calcium levels and the response to external oxidants [65]. For example, a short-term reduction in KGDHC similar to that in the current studies enhances the ability to diminish oxidants, whereas a long-term reduction diminishes the ability to buffer external oxidants [65]. Thus, a therapy that initially promotes autophagy followed by activation of mitochondria by compounds such as benfotiamine or PGC-1a to increase healthy mitochondria may be beneficial.

## 11. KGDHC and Calcium Signaling

While the role of calcium in activating mitochondrial dehydrogenases such as KGDHC has been studied extensively [84,150,151], relatively few studies have examined the role of KGDHC in regulating cellular calcium signaling. Considerable evidence suggests a direct connection of the endoplasmic reticulum (ER) to mitochondrial calcium through mitochondrial-associated particles. Evidence suggests some calcium released from the ER has a link to the mitochondria that differs from the whole cytosolic pool. ER-mitochondria-associated membranes likely mediate the interactions [150,152].

A highly replicable change in AD cells is enhanced IP3-releasable calcium from the ER. This was first reported in fibroblasts from patients with non-genetic forms of AD [153] and was replicated in fibroblasts from multiple other cell lines from control and AD patients [154], as well as in fibroblast and neurons from mice bearing AD-causing gene mutations [155]. We postulate that the change in the non-genetic forms of AD is related to diminished KGDHC. To test this possibility, the activity of the KGDHC was diminished acutely (minutes), long-term (days), or chronically (weeks) [65]. Acute inhibition of KGDHC produces effects on calcium opposite to those in AD, while the chronic or long-term inhibition of KGDHC mimics the AD-related changes in calcium. Thus, the results are consistent with the suggestion that long-term or chronic inhibition of KGDHC leads to AD-like changes in calcium regulation. The lack of change in the mitochondrial membrane potential suggests the effect is not likely due to energy failure [64,65]. Subsequent studies tested whether the relationship was true in human iPSC-derived neurons. Inhibition of KGDHC for one or twenty-four hours increased the ER releasable calcium stored in human neurons by 69% and 144%, respectively. The consequences are enzyme-specific since inhibiting PDHC did not diminish the ER-releasable calcium stores. The link of KGDHC to ER- releasable calcium stores is also cell type-specific since the interaction does not occur in iPSC or neural stem cells [64].

The mechanism for the association of KGDHC to calcium is likely related to the ability of KGDHC to alter multiple protein cellular signals (see Section 10). The calcium dysregulations seen in AD models are novel and pathogenic changes to fundamental calcium signaling patterns [156]. An increased number of ryanodine receptors appears important in mice with mutated presenilin. Increased ROS expression in neurons contributes to increased calcium release, synaptic decline, and increased RyR2 expression seen in the hippocampus and cortex. This increased IP_3_ receptor-evoked calcium release may involve the increase of stromal interaction molecule 1 (STIM1) expression and the decrease of STIM2 expression in the soma. These STIMs translocate to plasma membrane–ER contact sites, where they bind to Orai1 channels and open a Ca^2+^ influx pathway to slowly replenish ER Ca^2+^ stores using extracellular Ca^2+^ [157,158]). Thus, reduced KGDHC may alter the movement of these proteins. New mechanisms that target proteins to intracellular organelles are being discovered.

## 12. Conclusions

KGDHC is rate limiting for energy production by the TCA cycle and new research demonstrates the exciting role KGDHC plays in the cell including the response to oxidative stress, cell signaling, post-translational modification, epigenetics, and diverse metabolic pathways as well as the surprisingly specific signal changes. These intriguing results support the hypothesis that thiamine-dependent enzymes like KGDHC are important therapeutic targets to treat diseases from neurodegeneration to cancer.

## Figures and Tables

**Figure 1 ijms-23-12403-f001:**
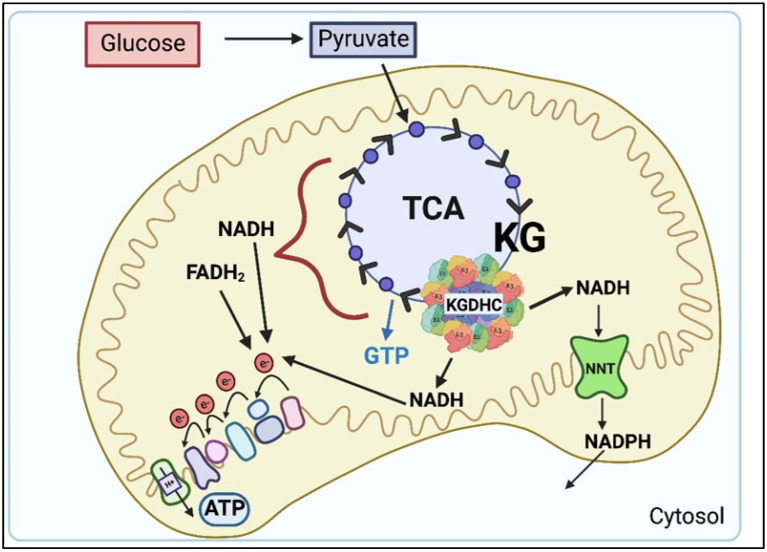
Overview of the traditional roles of α-ketoglutarate dehydrogenase complex (KGDHC). The tricarboxylic acid (TCA) cycle oxidizes pyruvate to produce NADH, FADH2 and GTP. Generally, the electrons from NADH and FADH2 are passed through a series of protein complexes to produce large amounts of ATP. The α-ketoglutarate dehydrogenase complex (KGDHC) is rate limiting for the TCA cycle. The NADH that is produced by KGDHC can be directed to nicotinamide nucleotide transhydrogenase (NNT) for conversion to NADPH, which is critical for oxidative carboxylation and maintaining cellular redox state.

**Figure 2 ijms-23-12403-f002:**
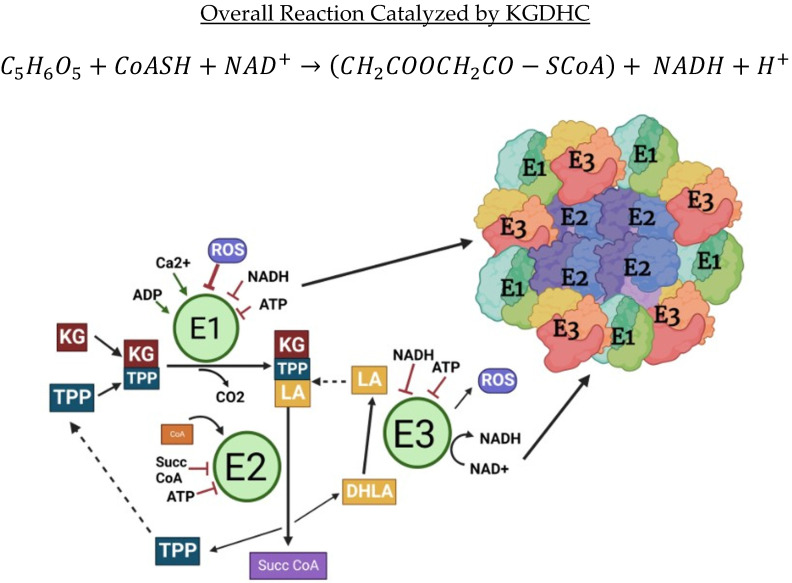
Structure and function of the α-ketoglutarate dehydrogenase complex (KGDHC). KGDHC is made up of three subunits. E1k, E2k, and E3. The core of KGDHC is com-posed of E2k proteins surrounded by E1k and E3 subunits forming an octahedral cube. This complex catalyzes the conversion of α-ketoglutarate (KG) to succinyl-CoA. KG combines with thiamine pyrophosphate (TPP) to enter the E1 subunit. This complex interacts with lipoic acid and E2k produces succinyl-CoA. Lipoic acid is reduced to dihydrolipoic acid (DHLA), which is oxidized by the E3 subunit to lipoic acid. Each subunit of KGDHC is uniquely regulated. Further, the E3 subunit produces ROS and the E1 subunit is inhibited by reactive oxygen species (ROS).

**Figure 3 ijms-23-12403-f003:**
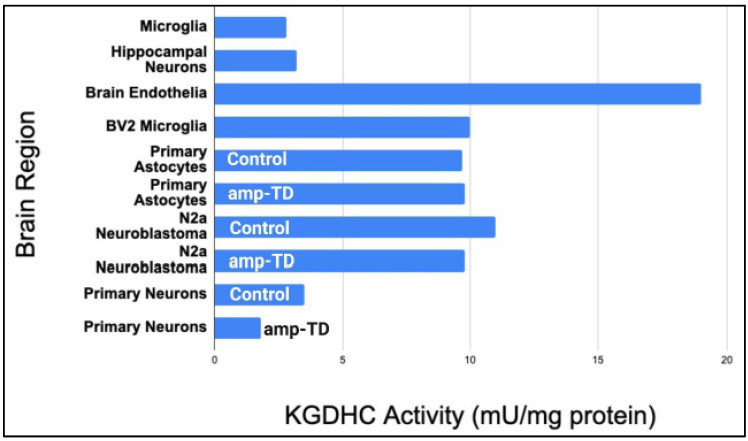
Activities of α-ketoglutarate dehydrogenase complex (KGDHC) and the response to thiamine deficiency (TD) vary between types of brain cells. Brain endothelial cells have the highest activity of KGDHC, while the hippocampal neurons have the lowest activity. KGDHC activity in neurons was the most responsive to TD produced by the thiamine transport inhibitor amprolium.

**Figure 4 ijms-23-12403-f004:**
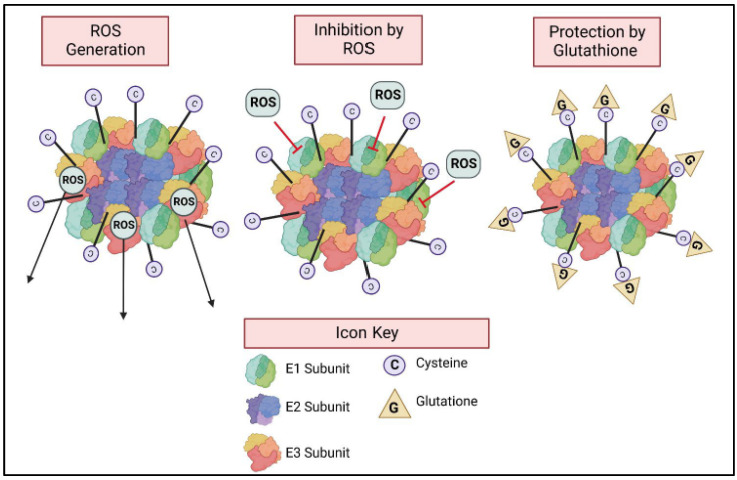
α-ketoglutarate dehydrogenase complex (KGDHC) and cell signaling by reactive oxygen species (ROS). The E3 subunit of KGDHC produces ROS. When KGDHC is active, ROS inhibits the E1 subunit and inhibits the overall activity of the complex. Glutathionylation protects KGDHC against ROS. Glutathione binding to the cysteine residues of KGDHC inactivates KGDHC and prevents ROS from damaging the complex.

**Figure 5 ijms-23-12403-f005:**
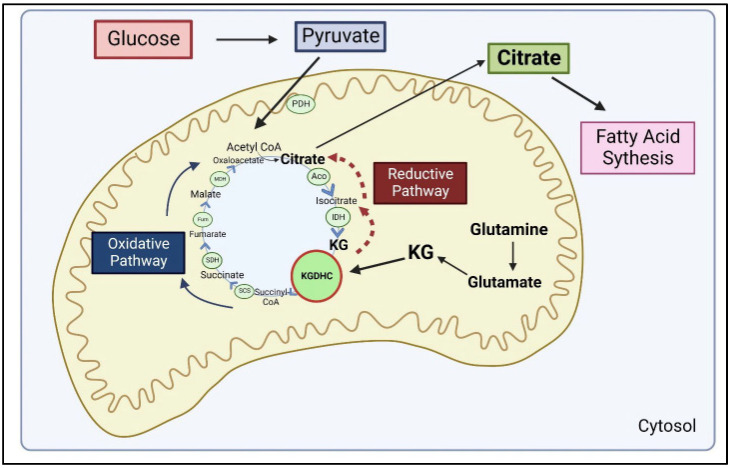
Metabolic plasticity of the α-ketoglutarate dehydrogenase complex (KGDHC). Glucose entering cells is converted to pyruvate which enters the inner mitochondrial matrix. The pyruvate dehydrogenase complex (PDH) oxidizes the pyruvate to acetyl CoA. The tricarboxylic acid (TCA) cycle begins when oxaloacetate combines with acetyl CoA to form citrate which is converted to isocitrate (by aconitase) and then to α-ketoglutarate (KG) by isocitrate dehydrogenase (IDH). KGDHC oxidizes KG to produce succinyl CoA which is converted to succinate by succinyl CoA synthetase. Succinate is converted to fumarate (by succinate dehydrogenase, SDH) then to malate (by fumarase, fum) and finally to oxaloacetate (by malate dehydrogenase, MDH). KGDHC is rate limiting so if it is inhibited, the oxidative pathway is impaired. The inhibition of KGDHC creates a blockage in the TCA cycle, forcing the reductive pathway to occur, which leads to higher amounts of citrate. This citrate is exported out of the mitochondria into the cytosol where it is used for fatty acid synthesis which is necessary for cell proliferation and growth. KG is also produced from glutaminolysis which can enter either the reductive or oxidative pathways of the TCA cycle depending on the activation of KGDHC.

**Figure 6 ijms-23-12403-f006:**
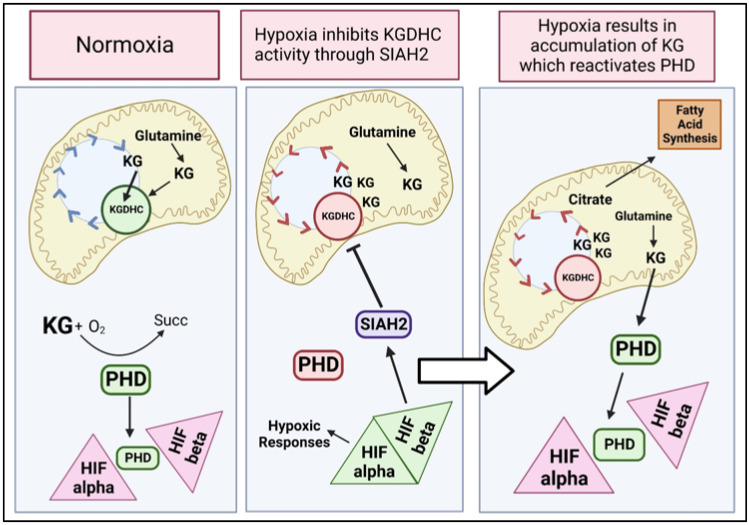
α-ketoglutarate dehydrogenase complex (KGDHC) interactions with HIF (hypoxia inducible factor). Normoxia. The tricarboxylic acid (TCA) cycle, α-ketoglutarate (KG) is converted to succinyl-CoA using the KGDHC enzyme complex. In the cytosol, proline hydroxylases (PHDs) are activated because there is plenty of KG and oxygen present. Once active, PHDs destabilize HIF.

**Figure 7 ijms-23-12403-f007:**
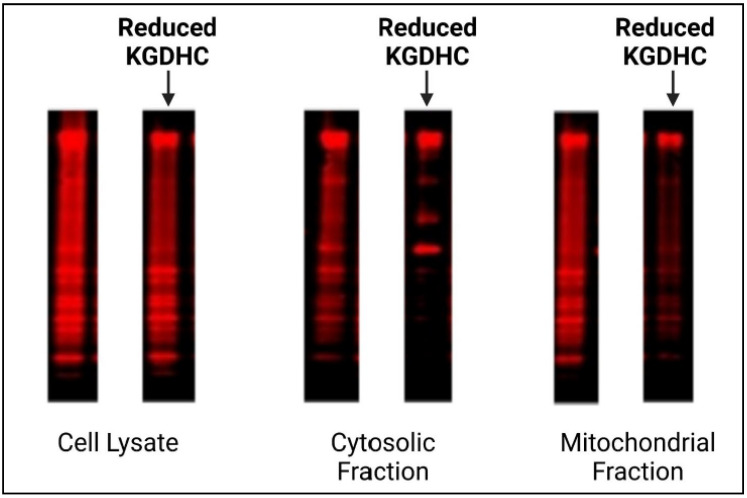
Inhibition of α-ketoglutarate dehydrogenase (KGDHC) reduces succinylation. Cultured neurons were incubated with or without a specific KGDHC inhibitor. The Westerns were probed with an anti-succinylation antibody. The red bands represent hundreds of succinylated proteins. In each group, the cells loaded on the lane on the right are from the KGDHC inhibited group. The blot shows the striking reduction in succinylation when KGDHC is reduced, especially in the mitochondrial fractions, demonstrating KGDHC’s role in cellular succinylation.

**Figure 8 ijms-23-12403-f008:**
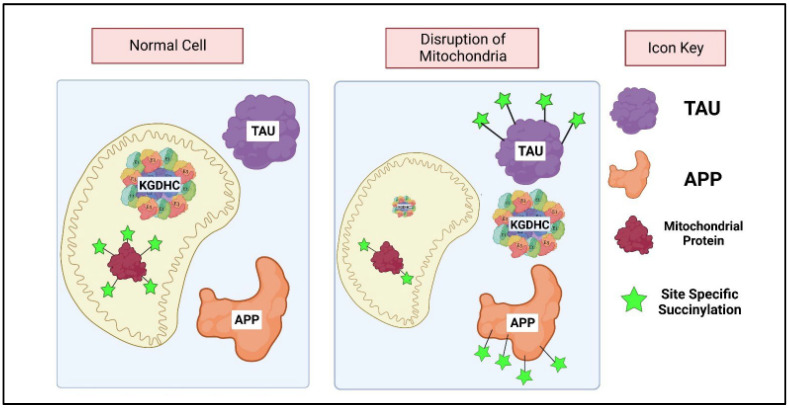
α-ketoglutarate dehydrogenase (KGDHC) and signaling by post translational modification through succinylation. In a normal functioning cell, KGDHC is present in the mitochondria and the E2 subunit of KGDHC is a succinyltrasferase. Mild metabolic disruption of the mitochondria causes KGDHC to leave the mitochondria and enter the cytosol where it succinylates cytosolic proteins such as amyloid precursor protein (APP) and Tau. KGDHC specifically succinylates the K612 part of APP (alpha secretase) and the K311 part of Tau (PHF 6 sequence). The succinylation of APP lead to increased amyloid β production, protein aggregation and formation of plaques. Succinylation of Tau blocks binding to microtubules, promotes aggregation and tangle formation. Both plaques and tangles play important roles in many neurological diseases, supporting the suggestion succinylation a key target for therapeutic treatments.

**Figure 9 ijms-23-12403-f009:**
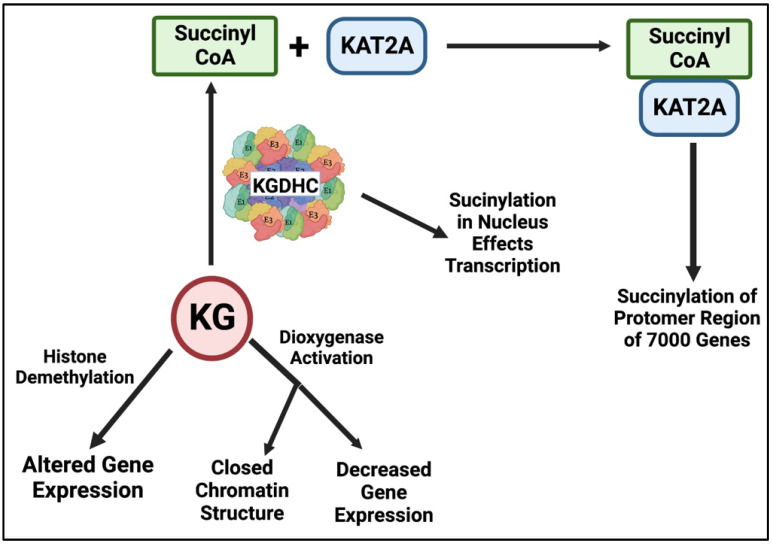
α-ketoglutarate dehydrogenase (KGDHC) and ketoglutarate (KG) in the regulation of transcription. KGDHC mediated succinylation can regulate gene expression by direct succinylation or through the lysine acetyltransferase (KAT2). KG can regulate histone demethylation or dioxygenase activation.

**Figure 10 ijms-23-12403-f010:**
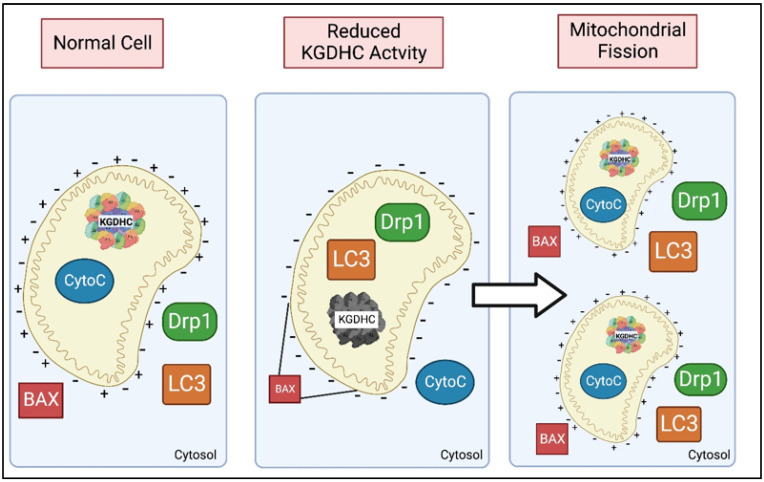
α-ketoglutarate dehydrogenase (KGDHC) and cellular protein signaling. Under normal conditions, Dynamin-related Protein 1 (Drp1) is present in the cytosol and cytochrome C is localized to the mitochondria. Inhibition of KGDHC results in Drp1 entering the mitochondria and cytochrome C translocating into the cytosol. The presence of Drp1 inside the mitochondria leads to the fission of the mitochondria. For cytochrome C to leave the mitochondria, Bcl-2 Associated X-protein Bax needs to be bound, but Bax needs a certain amount of mitochondrial membrane length to bind. Therefore, the fission of the mitochondria leads to less cytochrome C release because Bax has less room to bind.

**Table 1 ijms-23-12403-t001:** Diseases related to disruption of KGDHC.

Name	Disease Characteristics	Measurements	Sources
Infantile lactic acidosis	Extreme psychomotor retardation	Deficiency and mutations in *DLD*	[30,31]
Psychomotor retardation in childhood	Mental and physical slowing down of functioning	Disruption of E2k function	[32,33]
Intermittent neuropsychiatric disease with ataxia	Children have attention deficit disorders as well as mild ataxia, lack of coordination and some hypotonic weakness.	Deficiency in E3- single G229C mutation in *DLD*	[34]
Friedreich’s ataxia	Damage to the long tracts in the spinal cord and develop ataxia	A GAA repeats causes a defect in the *FRDA* gene as well as deficiency in the E3 subunit	[35,36,37,38]
Parkinson’s disease	Secondary motor function is impaired and a decline in intelligence is seen	Dysfunction of the DLST reduced KGDHC activity.	[39,40,41]
Huntington’s disease	Movement disorder followed by cognitive deficits.	Reduces KGDHC activity	[42]
Alzheimer’s disease	Loss of memory and cognitive function	KGDHC activity is reduced in the AD brain by 30% to 90%.	[36,41,42,43,44,45]
Progressive supranuclear palsy	Postural instability and cognitive impairment	50–60% reduction in KGDHC in superior frontal cortex and cerebellum	[46,47]
Spinal cord injury	Injury behavior similar to depression and cognitive decline (Animal model study)	KGDHC activity is decreased by 90% in the spinal cord an 30% in the cerebral cortex.	[48]
Head trauma	Injury to the skull, scalp, or brain (Animal model study)	Decreased KGDHC activity	[49]
Wernicke Korsakoff Syndrome	Severe memory deficits.	Decreased KGDHC	[43]

**Table 2 ijms-23-12403-t002:** Summary of primary changes caused by partial reduction of KGDHC.

Partial Reduction of KGDHC In Vivo	Partial Reduction of KGDHC In Vitro
Diminishes neurogenesis [63]	AD-like changes in calcium [64,65,66]
Reduces ability of the brain to adapt [60,67]	Reduced B cell activation [68]
Exacerbates plaque formation [69]	Selective cell and region death [70]
Causes hyper-phosphorylation of tau [69]	Increased MDH like AD [23]
Causes oxidative stress [71,72]	Increased autophagy/mitophagy, fission [73]
Reduces memory [74]	Abnormal oxidant buffering like AD [59]
Selective neuron loss [75,76,77]	Regulates GTP concentration [78]
Altered succinylation [79]	Altered succinylation [80]

## Data Availability

Not applicable.

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
