# Peer review of "The α-Ketoglutarate Dehydrogenase Complex as a Hub of Plasticity in Neurodegeneration and Regeneration"

_ijms, 2022, doi:10.3390/ijms232012403_

Round 1

Reviewer 1 Report

Dear Authors,

The alpha-ketoglutarate dehydrogenase complex (KGDHC) is key in regulating the energy production of cells, its reduced activity is observed in many neurodegenerative diseases, therefore understanding the decline will have a large translational and potential therapeutic impact.

In this manuscript, the authors demonstrate the multiple roles of KGDHC, including the response to oxidative stress, cell signaling, post-translational modification, epigenetics, and diverse metabolic pathways, as well as surprisingly specific signaling changes. Based on these, thiamine-dependent enzymes such as KGDHC may be important therapeutic targets.

The topic is timely and may attract much attention. The study is well-conducted and designed; however, I have some suggestions to improve this paper.

1. References:

In general, I recommend authors use more references to back their claims, especially in the Background of the article, which I believe is lacking. Thus, I recommend the authors attempt to deepen the subject of their article, as the bibliography is too concise. Nonetheless, in my opinion, less than 150-200 articles for a review paper are insufficient. Currently, authors cite only 103 papers, and in my opinion, they should cite more than it.

2. Figures:

The figures must be understandable by themselves, a short description is essential. In addition, the abbreviations used in the figures should be explained in the figure legends.

The quality of the figures also need to be improved, the text hard to read, and pixelated in some places (e.g.: Figure 6, 7).

3. Language proofreading is highly recommended. The are several formal and grammar errors.

4. Formal errors:

Abbreviations:

alpha ketoglutarate dehydrogenase complex/α-ketoglutarate dehydrogenase complex (KGDHC)- please unify the abbreviation

KGDHC (Line 49-50, 85), KG (Line 105, 107), ROS (Line 58, 114, 231) - the complete form of the abbreviation appears in several places.

The complete form of E2K appears only in Line 113, but until then, it is used in several places.

Space errors:

Space error occurs in several places. Please correct them.

(e.g.:Line 256 (NNT)(Figure1) - no space, Line 141 have  best - double space)

Typing errors:

Typing errors occurs in several places. Please correct them.

(e.g.: Line 520 he authors - missing T)

Font size errors:

Table 2 - last line, 2nd column (Injury to the skull..)-  larger font size, than others.

Recommendation of revision: major

Author Response

We thank the reviewer for his/her thoughtful comments.  Responding to them helped us improve the manuscript considerably. The whole manuscript was modified so extensively that the copy marked with MS Word tracker is not helpful. Thus, when appropriate we refer to the line numbers in the new version

Our comments are inserted in red between the reviewer’s concerns.

Reviewer #1

Manuscript ID

Comments and Suggestions for Authors

Dear Authors,

The alpha-ketoglutarate dehydrogenase complex (KGDHC) is key in regulating the energy production of cells, its reduced activity is observed in many neurodegenerative diseases, therefore understanding the decline will have a large translational and potential therapeutic impact.

In this manuscript, the authors demonstrate the multiple roles of KGDHC, including the response to oxidative stress, cell signaling, post-translational modification, epigenetics, and diverse metabolic pathways, as well as surprisingly specific signaling changes. Based on these, thiamine-dependent enzymes such as KGDHC may be important therapeutic targets.

The topic is timely and may attract much attention. The study is well-conducted and designed; however, I have some suggestions to improve this paper.

  1. References:

In general, I recommend authors use more references to back their claims, especially in the Background of the article, which I believe is lacking. Thus, I recommend the authors attempt to deepen the subject of their article, as the bibliography is too concise. Nonetheless, in my opinion, less than 150-200 articles for a review paper are insufficient. Currently, authors cite only 103 papers, and in my opinion, they should cite more than it.

We agree with the reviewers.  The original manuscript contained 103 references. We increased that to 164- references. Most additions were in the background section and in the section on KGDHC in Alzheimer’s disease. Adding these references mandated that much of the paper be re-written. Nevertheless, we do not regard this as an exhaustive summary of all aspects of the fundamental chemistry of KGDHC. We are trying to present enough so that the reader can understand the implications of the changes in KGDHC in neurodegeneration.

  1. Figures:

The figures must be understandable by themselves, a short description is essential. In addition, the abbreviations used in the figures should be explained in the figure legends.

We added legends to the figures to make them understandable on their own. The new legends also define the   abbreviations on the figure. 

The quality of the figures also need to be improved, the text hard to read, and pixelated in some places (e.g.: Figure 6, 7).

We re-evaluated each figure to improve the readability and remove types. Many of the figures were modified. Figures 6 and 7 were improperly imported into the text.

  1. Language proofreadingis highly recommended. The are several formal and grammar errors.

The document was extensively re-written to address this concern. We carefully revised document for formatting and grammar errors.

  1. Formal errors:

Abbreviations:

alpha ketoglutarate dehydrogenase complex/α-ketoglutarate dehydrogenase complex (KGDHC)- please unify the abbreviation

KGDHC (Line 49-50, 85), KG (Line 105, 107), ROS (Line 58, 114, 231) - the complete form of the abbreviation appears in several places.

We made sure that each time we use KGDHC we refer to the α-ketoglutarate dehydrogenase complex.

We made sure that each time we use KGDH we refer to the first enzyme of the complex -α-ketoglutarate dehydrogenase complex

We have changed all alphas to the symbol α

The complete form of E2K appears only in Line 113, but until then, it is used in several places.

We now use the whole name the first time E2k is introduced in the text (line 122) We have searched the whole document to make sure that each abbreviation is defined the first time it is introduced.

Space errors:

Space error occurs in several places. Please correct them.

(e.g.:Line 256 (NNT)(Figure1) - no space, Line 141 have  best - double space)

We made sure that appropriate spaces were inserted. Our understanding is that only single spaces should be used.  We searched the whole document double spaces, for spaces before a ( or [.

Typing errors:

 We have carefully checked for formatting errors as we re-wrote the document.

Typing errors occurs in several places. Please correct them.

(e.g.: Line 520 he authors - missing T)

We have carefully checked for typos as we re-wrote the document.

Font size errors:

Table 2 - last line, 2nd column (Injury to the skull..)-  larger font size, than others.

We searched the whole document for font size errors.

Recommendation of revision: major

Reviewer 2 Report

This manuscript presents an interesting review of the Keto-glutarate dehydrogenase complex and its involvement in cell regulation.

The subject is not my speciality. But I learned or refreshed a number of concepts in reading it.

However as a chemist (and biochemist) I wonder if a the equations being mentioned in the text should not be fully explicited in this review : You could number them and then recall their number when needed in the text or legends of figures. I know, these equations  are present in any Biochemistry textbook, but the reader does not always remembers the formula of all the metabolites cited and their formation.

I think it would improve the readability for non specialists.

The paper seems quite well written. I found it interesting.

Thus I would suggest minor revision as explained above.

Author Response

We thank the reviewer for his/her thoughtful comments.  Responding to them helped us improve the manuscript considerably. The whole manuscript was modified so extensively that the copy marked with MS Word tracker is not helpful. Thus, when appropriate we refer to the line numbers in the new version

This manuscript presents an interesting review of the Keto-glutarate dehydrogenase complex and its involvement in cell regulation.

The subject is not my speciality. But I learned or refreshed a number of concepts in reading it.

However as a chemist (and biochemist) I wonder if a the equations being mentioned in the text should not be fully explicited in this review : You could number them and then recall their number when needed in the text or legends of figures. I know, these equations  are present in any Biochemistry textbook, but the reader does not always remembers the formula of all the metabolites cited and their formation.

I think it would improve the readability for non specialists.

 We have added the equations as suggested. These insertions helped make the manuscript much easier for the non-chemist to understand the beauty of KGDHC

Line 120, line 143, line 164-165, line 187

The paper seems quite well written. I found it interesting.

 Thank you. We hope the new version will be even more interesting.

Thus, I would suggest minor revision as explained above.

Reviewer 3 Report

The manuscript "The α-Ketoglutarate Dehydrogenase Complex as a Hub of Plasticity in Neurodegeneration and Regeneration" reviews the role of the KGDHC, which is not limited to its metabolic function. The review summarizes the data on this multienzyme complex, its localization and connection to other proteins and neurodegeneration or cancer. The authors are experts in the field, and their review should be a good contribution to the Special Issue "Role of the 2-Oxo Acid Dehydrogenase Complexes in Metabolism: Genetics, Structure and Function".

However, the current version of manuscript is far from a variant, which could be approved for publication. My comments to the text are summarized below:  

1. The part of section 1, corresponding to Fig.1 is too long. Most of the data from Fig.1 is well known, so corresponding text can be shortened.

2. The part of section 2 (Fig. 2) about KGDHC regulators should be supported by references and be more specific, because such regulation differs between species. A table similar to Table 2 (with references for each ligand) would be much more interesting here, while current version of Table 1 can be moved to supplementary, omitted or added with the information about ligands and references. Data on protein-protein interactions and regulation of KGDHC function could be added here together with the discussed role of small molecules.

3. Section 3. Due to the similarity of sentences in lines 120 and 140 or the construction of the sentences in lines 136-138, or lines 155-157, or multiple typos in the section and table, the section seems to be raw, although relevant.
The discussed KGDHL is not added to the current table 1. The authors should discuss, how the two isoenzymes are arranged within the complex.

4. Section 4. The reference 39/40 is duplicated. Thus, the paragraph corresponding to Fig. 3 and the figure itself corresponds to a single paper. The Fig.3 in the review is much less informative than the figure 3 from the original paper (https://doi.org/10.1046/j.1471-4159.2000.0740114.x) The use of neuroblastoma here looks especially odd, regarding the comment below the figure 3. This paragraph needs independent data to be incorporated.

In addition, the discussion of thiamine deficiency added to the paragraph (l. 190-202) should be arranged as a separate paragraph. In principle the figure could provide an ability to compare the effect of TD in the cells. This paragraph and the figure need significant correction.

5. Section 5. So, what is your hypothesis for the link between ER Ca store and KGDHC, if not the energy failure? The manuscript would benefit from a scheme for the proposed mechanism.

6. The mechanisms, described in section 6 and Fig. 4 suggest also lipoate damage as a potential mechanism for regulation by ROS. Can you address this point too?

7. A brief comparison of the data with KGDHC inhibitor to independent data from mutant cells/organism is needed in section 9. In addition, other PTM and their relation to succinylation could be mentioned in the end of this section.

8. The end of section 9 is in good agreement with the section 10, so it could even fit better to the section 10. What's more important, the section needs a schematic image of the discussed role of KG and KGDHC in methylation of DNA and proteins, succinylation, KAT2A, and maybe HIF and p53 discussed in the paper, or different pathways like apoptosis and autophagy (discussed in section 10).

Minor comments:

The graphical abstract could be in a better agreement with the text of the abstract. It's not clear, whether a mitochondrion or a cell is depicted. Addition of nucleus would improve the scheme, since it is directly discussed in the manuscript together with involvement of KGDHC in the cytosolic and nuclear processes.

line 58. A couple of references supporting ROS production and regulation are needed here.

lines 78-82. The end of section 1 needs references.

Typos, such as "feedback look" (l. 331), "is move sensitive" (l. 472) or duplications (l. 430 "at specific sites (62 ± 7%) at specific sites") should be corrected.

Author Response

We thank the reviewer for his/her thoughtful comments.  Responding to them helped us improve the manuscript considerably. The whole manuscript was modified so extensively that the copy marked with MS Word tracker is not helpful. Thus, when appropriate we refer to the line numbers in the new version

The reviewer’s comments are outstanding, and we have carefully addressed each concern. We have inserted our comments in red between the reviewers’ comments.

The reviewer’s comments are in black and our responses inserted in red.

REVIEWER 3

The manuscript "The α-Ketoglutarate Dehydrogenase Complex as a Hub of Plasticity in Neurodegeneration and Regeneration" reviews the role of the KGDHC, which is not limited to its metabolic function. The review summarizes the data on this multienzyme complex, its localization and connection to other proteins and neurodegeneration or cancer. The authors are experts in the field, and their review should be a good contribution to the Special Issue "Role of the 2-Oxo Acid Dehydrogenase Complexes in Metabolism: Genetics, Structure and Function".

However, the current version of manuscript is far from a variant, which could be approved for publication. My comments to the text are summarized below:  

  1. The part of section 1, corresponding to Fig.1 is too long. Most of the data from Fig.1 is well known, so corresponding text can be shortened.

We reduced the text associated with Figure 1 from 456 words to 175 words. Lines 103-115.

However, we added an overview so that a reader is aware of the goals of the review and can appreciate the rest of the review. (Lines 77-102). The review is intended for a wide audience so some overview is necessary. On the other hand, whole reviews have been written on each aspect covered.

  1. The part of section 2 (Fig. 2) about KGDHC regulators should be supported by references and be more specific, because such regulation differs between species. A table similar to Table 2 (with references for each ligand) would be much more interesting here, while current version of Table 1 can be moved to supplementary, omitted or added with the information about ligands and references. Data on protein-protein interactions and regulation of KGDHC function could be added here together with the discussed role of small molecules.

We did not mention all the species difference in regulation.  Our intent was not to review all the regulatory aspects of KGDHC but to present an overview so the reader could appreciate the role in neurodegeneration. Although in the long term these may have a bearing, they are not directly relevant to the subsequent review. That does not have bearing on the overall theme of the review. We created a supplementary Table 1  (line 1221) with the more detailed chemical/biochemical information. A detailed discussion of these features could be a review on its own.

  1. Section 3. Due to the similarity of sentences in lines 120 and 140 or the construction of the sentences in lines 136-138, or lines 155-157, or multiple typos in the section and table, the section seems to be raw, although relevant.

We agree this section of the review was embarrassingly “raw”. We have completely rewritten this section.

The discussed KGDHL is not added to the current table 1. The authors should discuss, how the two isoenzymes are arranged within the complex.

We have added more text about KGDHL (Lines 160-166). We also added KGDHL to the supplementary table. To our knowledge the arrangement of KGDHL in the complex has not been established.

  1. Section 4. The reference 39/40 is duplicated. Thus, the paragraph corresponding to Fig. 3 and the figure itself corresponds to a single paper. The Fig.3 in the review is much less informative than the figure 3 from the original paper (https://doi.org/10.1046/j.1471-4159.2000.0740114.x) The use of neuroblastoma here looks especially odd, regarding the comment below the figure 3. This paragraph needs independent data to be incorporated.

The whole manuscript has been reviewed for duplicate references. We revised figure 3 to make it clear. We cannot add independent data because we do not know of any comparable results where activities have been measured in one cell type in one lab.  We added the following. Mitochondrial heterogeneity in various cell type in other organs has been reviewed [1]. While the relative activities as compared to protein levels are very important in understanding the role of KGDHC in the pathophysiology of disease, the studies described in Figure 3 have important limitations: 1. They have not been replicated. 2. Cells in culture are very different than in a tissue. 3. The figure includes transformed cells which alters metabolism. (lines 298-302).

In addition, the discussion of thiamine deficiency added to the paragraph (l. 190-202) should be arranged as a separate paragraph. In principle the figure could provide an ability to compare the effect of TD in the cells. This paragraph and the figure need significant correction.

We moved the thiamine deficiency to a separate paragraph as suggested (lines 304-311). We also add the thiamine deficiency to Figure 3.

  1. Section 5. So, what is your hypothesis for the link between ER Ca store and KGDHC, if not the energy failure? The manuscript would benefit from a sceme for the proposed mechanism.

We added the following sentences (Lines 640-651). This is very speculative, so we did not add a figure.

  1. The mechanisms, described in section 6 and Fig. 4 suggest also lipoate damage as a potential mechanism for regulation by ROS. Can you address this point too?

Sentences on the role of lipoate oxidation state have been added into section 6 (Lines 319-350).

  1. A brief comparison of the data with KGDHC inhibitor to independent data from mutant cells/organism is needed in section 9. In addition, other PTM and their relation to succinylation could be mentioned in the end of this section.

We added comments about the degree of inhibition in our cell lines by our method that measures in situ KGDHC activity. The experiments in yeast did not measure activities so direct comparisons are not possible. (Lines 470-480)

We added discussion about comparison with various post-translational modifications (lines 507-515).

  1. The end of section 9 is in good agreement with the section 10, so it could even fit better to the section 10. What's more important, the section needs a schematic image of the discussed role of KG and KGDHC in methylation of DNA and proteins, succinylation, KAT2A, and maybe HIF and p53 discussed in the paper, or different pathways like apoptosis and autophagy (discussed in section 10).

 We agree we moved the end of section 9 into section 10. We added a cartoon for the gene regulation (Figure 9) and reworded the P53 sentences. Lines 558-564.

We added a new cartoon on succinylation in the nucleus. (Figure 9)

Minor comments:

The graphical abstract could be in a better agreement with the text of the abstract. It's not clear, whether a mitochondrion or a cell is depicted. Addition of nucleus would improve the scheme, since it is directly discussed in the manuscript together with involvement of KGDHC in the cytosolic and nuclear processes.

We made the graphical abstract simpler not more complicated. We omitted the cell structure and just list functions. We think summarizes everything very well, and is far less confusing

line 58. A couple of references supporting ROS production and regulation are needed here.

We now have many references of this possibility. Lines 314-350)

lines 78-82. The end of section 1 needs references.

This whole section was completely rewritten (Lines 76-118) and supplementary table 1 (line 1227)

Typos, such as "feedback look" (l. 331), "is move sensitive" (l. 472) or duplications (l. 430 "at specific sites (62 ± 7%) at specific sites") should be corrected.

We have re-read the re-written manuscript many times for such errors.

Round 2

Reviewer 1 Report

Dear Authors, 

I appreciate that the authors have taken my considerations into account, and all my concerns have been addressed.

Author Response

No changes were requested.

Reviewer 3 Report

This version of the manuscript is significantly improved compared to the previous one. Still several points should be addressed. Below I've listed the corrections which are necessary to be included.

1. The graphical abstract should contain labels of unified style. The "Rate Limiting Enzyme of the TCA Cycle" differs from the other functions by its style. Besides, no such function is indicated in the abstract, while no indication of "regulates the cellular responses to hypoxia" function is made in the graphical abstract.

2. Line 94: "is directed" (about NADH) shouldn't be so critical, because it depends on e.g. NNT expression as follows from Ref 5. So, "can be directed" should be written. The same should be corrected in the figure caption. This is a very important moment, as a great difference between even skeletal and cardiac muscle exists (Ref. 5). I can't but cite a sentence from the paper: "...experiments pinpointing α-KGDH as a major source of ROS were performed in skeletal muscle or brain mitochondria [303438], where NNT activity is substantially lower than in the heart [28]."
So, the flow on NADH from KGDHC can also go to Complex I, which is not covered by Figure 1. Most likely, other NADH sources can also provide the substrate for NNT.

3. Figure 9 and its caption. KATA2 should be corrected to KAT2A.

4. line 566. "ATG" is used without description although it is not one gene.
